# Dietary Fat Effect on the Gut Microbiome, and Its Role in the Modulation of Gastrointestinal Disorders in Children with Autism Spectrum Disorder

**DOI:** 10.3390/nu13113818

**Published:** 2021-10-27

**Authors:** Monia Kittana, Asma Ahmadani, Farah Al Marzooq, Amita Attlee

**Affiliations:** 1Department of Nutrition and Health, College of Medicine and Health Sciences, United Arab Emirates University, Al Ain P.O. Box 15551, United Arab Emirates; 202090016@uaeu.ac.ae (M.K.); 201270079@uaeu.ac.ae (A.A.); 2Department of Medical Microbiology and Immunology, College of Medicine and Health Sciences, United Arab Emirates University, Al Ain P.O. Box 15551, United Arab Emirates; f.almarzooq@uaeu.ac.ae

**Keywords:** autism spectrum disorder, dietary fat, fatty acid, gastrointestinal disorder, gut microbiome

## Abstract

Children with autism spectrum disorder (ASD) report a higher frequency and severity of gastrointestinal disorders (GID) than typically developing (TD) children. GID-associated discomfort increases feelings of anxiety and frustration, contributing to the severity of ASD. Emerging evidence supports the biological intersection of neurodevelopment and microbiome, indicating the integral contribution of GM in the development and function of the nervous system, and mental health, and disease balance. Dysbiotic GM could be a contributing factor in the pathogenesis of GID in children with ASD. High-fat diets may modulate GM through accelerated growth of bile-tolerant bacteria, altered bacterial ratios, and reduced bacterial diversity, which may increase the risk of GID. Notably, saturated fatty acids are considered to have a pronounced effect on the increase of bile-tolerant bacteria and reduction in microbial diversity. Additionally, omega-3 exerts a favorable impact on GM and gut health due to its anti-inflammatory properties. Despite inconsistencies in the data elaborated in the review, the dietary fat composition, as part of an overall dietary intervention, plays a role in modulating GID, specifically in ASD, due to the altered microbiome profile. This review emphasizes the need to conduct future experimental studies investigating the effect of diets with varying fatty acid compositions on GID-specific microbiome profiles in children with ASD.

## 1. Introduction

Autism spectrum disorder (ASD) is a neurodevelopmental condition that is characterized by challenges in social skills and communication, along with the presence of restricted and repetitive behaviors [1]. A high rate of both mental and physical conditions is reported in individuals with ASD [2], including epilepsy, sleep disorders, anxiety, bipolar disorder, attention deficit hyperactivity disorder, and gastrointestinal disorders (GID) [3]. Moreover, the increasing worldwide prevalence of ASD [4], along with unmet health needs, contributes to a high disease burden in individuals with ASD [2]. Children with ASD commonly develop several gut-related comorbidities with gastrointestinal symptoms associated with increased gut epithelial barrier permeability, the decreased expression of brush border disaccharidases in the gut epithelium, and altered gut microbiome (GM) composition [5]. When untreated, GID can contribute to increased medical costs, trigger sleep disturbances, and increase social withdrawal and anxiety [6,7,8]. Additionally, GI symptoms may appear and manifest as behavioral changes [9]. The unpleasant feeling associated with GID, such as pain and discomfort, may lead to feelings of anxiety, frustration, and irritability [6,10], which may increase the risk of aggression and self-abuse [10], especially if children are unable to express this discomfort verbally, thus increasing the severity of ASD itself. 

GID in children with ASD are consistently reported to be of a higher frequency and severity than in typically developing (TD) children [10]. Various studies reported GID prevalence, ranging from 23% to 85%, with diarrhea and constipation being the most common [6,10]. In a recent review of 114 reports on ASD and GID, the authors reported the prevalence rate of 4.3–45.5% for constipation, 2.3–75.6% for diarrhea, and 4.2–96.8% for the presence of any GI symptoms [11]. This variety in GID frequency could be considered to be reflective of the heterogeneity of the ASD condition itself; however, there is at least a three times higher risk of GID in children with ASD in contrast to TD children, [6]. For example, Adams et al. (2011) reported that the prevalence of moderate–severe diarrhea and/or constipation was 63% in children with ASD compared to only 2% in TD children [10]. A higher prevalence of GID in female than male children with ASD has been reported [12,13]. Children with ASD tend to more commonly present GI symptoms early in life than TD children [14]. Further, the Norwegian Mother and Child Cohort Study showed that, in contrast to TD children, children with ASD had greater odds of having mother-reported constipation in the ages of 6–18 months and 18–36 months, and reported diarrhea in the ages of 18–36 months [15]. Several causes have been postulated for the higher prevalence rate of GID in children with ASD. For example, constipation may be attributed to a restricted diet usually low in fiber, side effects of medications, and/or irregular toileting behaviors due to developmental deficits [8,16,17]; however, not all of these factors were consistently associated with GID [18]. Diarrhea can also be caused due to the side effects of medications [7] and by other intestinal conditions, such as food allergies and/or intolerances [19]. Additionally, the GM could play a significant role [5].

The commensal bacteria of GM communicate with each other and with the gut epithelium of the host to maintain the gut homeostasis and improve the host immunity [20,21]. This communication is mediated by bacterial structural components and metabolites that act as signaling molecules for interaction with the gut epithelial cells as well as the cells of the immune system [21]. This interaction is essential for the maintenance of tissue homeostasis by a complex and coordinated set of innate and adaptive immune responses against different triggers such as food-derived antigens, metabolites, and pathogens [20]. GM is involved in gastrointestinal physiology, metabolism, nutrition, immune function, in addition to brain functions, including behavior of an individual [22]. Growing evidence suggests that the gut has a strong bidirectional communication with the brain, which is vital for maintaining brain functions and gut homeostasis. Microbiome–gut–brain axis mediates this communication between GM and the central nervous system (CNS). It plays a crucial role in several brain processes such as neuroinflammation, neurotransmission, the synthesis of neurotransmitters, and the regulation of complex behavioral functions such as anxiety, stress, and sociability [22]. For example, the intakes of prebiotics and probiotics and dietary polyphenols increase the production of phenolic acids from the microbial metabolism, exerting desirable effects in terms of neuroprotection [23]. GM is beneficial to the host at a healthy state, and its disruption (known as dysbiosis) can lead to several diseases [24]. It is hypothesized that dysbiosis contributes to the development of GI problems, which occur partly due to the disrupted communication between the microbiome, intestines, and brain [25]. An abnormal GM has been associated with several diseases, such as inflammatory bowel disease (IBD) [26], mood disorders [27], and ASD [28]. Several studies have shown alterations in the composition of the GM and metabolic products of the GM in children with ASD [29,30]. Various classes of bacteria have been found to be significantly different in children with ASD. For example, *Bacteroidales*, found significantly lower in children with ASD, have been suggested to affect brain function and development of neurological function, and the alternation of food intake has been demonstrated to improve neurodevelopment by interfering with GM composition [31]. Although specific bacterial strain-induced behavioral alterations in animal experiments have demonstrated the effect of GM in regulating certain processes of the CNS, such findings in humans scarce understand the underlying mechanisms [29]. 

The growing focus on the microbiome–gut–brain axis has encouraged many investigators to develop ways for the reversal of GM alterations as therapeutic interventions for alleviating ASD symptoms. Dietary patterns, antibiotic use, and infections are often associated with the alternations in GM composition and the loss of gut homeostasis, which have been implicated in the pathogenesis of gut as well as brain disorders. Probiotic therapy, for example, in managing GI dysfunction, has been shown as a promising approach, beneficially altering fecal microbiota, and reducing the severity of ASD symptoms in children [32]. Dietary manipulations for GID are considered an important part of the overall therapeutic plan. Dietary fat plays a significant role; however, in comparison with other nutrients, the relationship of dietary fat and the microbiome is more complicated and is not yet fully understood. Factors such as the dietary fat percentage and the fatty acid profile of the diet affect the GM community and the inflammatory response [33]. GM can be affected by several dietary factors. Treatments based on the microbiome modulation with prebiotic, probiotic, and dietary interventions may influence not only the gut, but also brain health and behavior.

Interestingly, significant differences are reported in the GM of children with ASD and TD [10], some of which are correlated with gastrointestinal health, ASD severity, or both [10,29]. Defining a specific microbiome profile in children with ASD and exploring whether it is related to GID would provide more of an insight on the etiology of GID and present new therapeutic alternatives for ASD. The aim of this review is to investigate the role of dietary fat in modulating GID, mainly through its effects on the GM balance and their byproducts. The article is divided into two main sections. The first section reviews the literature on GM composition and the association with behavior and GID, and the second section summarizes the evidence on the effect of dietary fat on GM composition and gut health. The evidence available in context to individuals/children with ASD is included.

## 2. Methods

A title/abstract-specific search was performed on PubMed. Google scholar was used as a secondary strategy for retrieving more articles, in addition to the reference list of the identified trials and reviews. The search included publications from any date up to August 2021. Two search strategies were employed: The first used autism-related terms (e.g., ‘Autism’, ‘Autism, infantile’, ‘Autistics disorder’) and keywords related to gut microbiome (e.g., gut microbiome, microbiota, or flora). The second used dietary-fats related terms (e.g., saturated fatty acids, monounsaturated fatty acids, polyunsaturated fatty acids, and omega-3 fatty acids (including EPA and DHA)) in addition to gut-microbiome related terms. All study designs, including review articles, were included in this review. Due to the lack of published articles on the effect of dietary fats in children/individuals with ASD, articles exploring its effect in different population groups and animal models were included to establish its association with GM. With that, it was compared to the altered GM composition in ASD.

## 3. GM and GID in Children with ASD

The human microbiome consists of the 10–100 trillion symbiotic microbial cells, the majority of which reside in the gut, making GM the largest microbiome in the human body [34]. Various techniques have been used to study GM, including culture, polymerase chain reaction (PCR), real-time PCR, and microarray [35]. Given the advancements in genomic techniques, next generation sequencing (NGS) by targeted 16s rRNA sequencing, or even shotgun metagenomic sequencing, were employed to determine the composition of GM; however, differences may arise due to sample handling; DNA extraction; sequencing of different regions of DNA; varying coverage achieved by PCR primers; sequencing depth; and bioinformatic and statistical methods used for analysis [36]. Nevertheless, NGS expands our ability to understand the human microbiome and its association with various diseases, including ASD. 

Generally, the heterogeneous GM colonizing the intestine, particularly the large intestine, is an essential component for supporting nutrition and metabolism [37], and forms a shield-like barrier to protect itself from pathogens and to support the gut immunity and overall intestinal integrity [37,38]. The essential GM, also known as normal flora, consists mainly of four major phyla (*Bacteroidetes, Firmicutes, Proteobacteria*, and *Actinobacteria*), in addition to two minor phyla (*Verrucomicrobia* and *Fusobacteria*) [24]. These phyla include several genera, mainly *Bacteroides*, *Bifidobacterium*, *Lactobacilli*, *Streptococci* (aerobic and anaerobic), *Clostridia*, *Eubacteria*, and *Enterobacterium* [38,39,40]. 

In a healthy gut, the *Firmicutes* predominate the small intestines, and the *Bacteroidetes* predominantly populate the colon [39]. Both *Bacteroidetes* and *Firmicutes* help in the metabolism of undigested food remains. Colonic bacteria including *Bacteroides, Roseburia, Bifidobacterium* and *Enterobacteriaceae* are involved in the fermentation of carbohydrates and indigestible oligosaccharides [41], which results in the synthesis of various short-chain fatty acids (SCFAs), including butyrate, propionate, and acetate [42]. These SCFAs play an important role in gut health as they exert anti-inflammatory, anti-tumorigenic, and immunomodulatory effects, preserving the intestinal barrier integrity [28,42,43]. Moreover, they are considered to be an energy supply for epithelial cells, specifically butyrate, that preserves the colonocytes function and homeostasis [42], supports normal bowel movements [44], and enhances intestinal barrier integrity [45]. Other bacteria such as *Bacteroides thetaiotaomicron* have been shown to induce greater expression of colipase, an enzyme required by pancreatic lipase for lipid digestion. *Lactobacillus plantarum* maintains intestinal barrier integrity and prevents the entry of gut bacteria, bacterial toxins, partially digested fats, and proteins into the bloodstream, thus preventing leaky gut [46].

Dysbiosis refers to the imbalance of microbial communities in a host [47]. It can occur as a result of losing beneficial microorganisms, increased growth of pathogenic microorganisms, or due to a loss of microbial diversity [47]. This can alter the levels of SCFAs, thereby contributing to the gut inflammatory process [43,48]. Dysbiosis can manifest as GID, including constipation, diarrhea, abdominal pain, bloating, and withholding [49], through its effect on immune dysregulation, altered signaling (e.g., altered serotonin synthesis), and intestinal barrier dysfunction [50]. Dysbiosis associated with specific diseases is characterized by unique compositional and functional changes in the microbiome [47]. 

Interestingly, the intestinal microbiome is capable of bidirectional interactions with the higher brain center via endocrine, neural, and immune system signaling [51]. This brain–gut–microbiome axis explains how changes in GM can be associated with both ASD and GID [49]. 

In dysbiosis, some pathogenic bacteria in the presence of a weakened immune system may lead to neurological disorders. Toxins produced by the bacteria may build up in the bloodstream, leading to confusion, delirium, and even a coma [46]. Specifically for ASD, significant differences are reported in the GM of children with ASD and TD children [10]. Strong evidence in animal models proves the relationship between the GM alterations and autism, whereby GM transplantation from human autistic donors into germ-free mice led to the induction of autistic behaviors in mice [52]. 

Both animal and clinical studies have demonstrated that modifying GM can treat ASD-related behaviors [49,52,53] by modulating the neural excitability and structuring the amygdala—the main hub—for regulating fear and social behaviors [54]. Moreover, GM influences the production and expression of neurotransmitters and neuromodulators [55,56], which activate or inhibit central neurons [57]. Members of the genera *Lactobacillus* and *Bifidobacterium* synthesize gamma-aminobutyric acid (GABA); *Lactobacillus* synthesizes acetylcholine; *Bacillus* and *Serratia* synthesize dopamine; *Candida, Streptococcus, Escherichia*, and *Enterococcus* synthesize serotonin; and *Escherichia, Bacillus* and *Saccharomyces* synthesize norepinephrine [58]. Alterations in these molecules mediate the effect of GM on brain development [59,60]. Animal studies have proven that the modulation of the GM can affect the manifestations of oxytocin, vasopressin, and brain-derived neurotrophic factor (BDNF), and improve the function of microglial cells, thus it can enhance behavior, learning, and memory in humans [59,61]. BDNF is a member of neurotrophins family, which regulate the function of nervous system [62]. Changes in BDNF can manifest as behavioral abnormalities, including anxiety-like behaviors [63]. The GM plays an interesting role in ASD, as it can influence both behavioral symptoms and ASD-related GID. This connection is highlighted in Figure 1.

Behavioral changes can also impact eating behaviors, which in turn contribute to GID, dysbiosis, and dysbiosis-related GID. Wallisch et al. found that children with ASD who were picky eaters were significantly more likely to have behavioral differences compared to non-picky eaters [64]. Further-more, picky eating was also significantly associated with increased odds of GI symptoms and sleep disturbances in children with ASD [12]. Sleep disturbances in turn increase the risk of lower scores in social cognition and communication, intellectual development, and the performance of daily living skills [12]. Eating behaviors can also impact GM due to the compromised diet quality. For example, it has been reported that children with ASD have a significantly lower fiber intake compared to TD children [65,66], and have a high preference towards sweet snacks and soft drinks [67,68,69]. Picky eating also affects GID risk. For example, higher food fussiness has been found to increase the risk of functional constipation, and functional constipation occurrence predicted a greater risk of picky eating in children, indicating the presence of a bi-directional relationship between food intake and GID, leading to the developing of a vicious cycle between these two factors [70]. Addressing these complications can prove to be more challenging in ASD, as the child’s food preferences, lack of acceptance of new foods, and lack of parental knowledge may pose limitations in adhering to nutritional advice [71]. 

Defining a specific GM composition for children with ASD is not yet possible [42]. However, some common observations have been reported. Most notably, children with ASD have a significantly decreased ratio of *Bacteriodetes*/*Firmicutes* phyla. *Bacteriodetes* is a phylum that includes 241 genera classified in 4 classes, including: *Bacteroidia*, *Flavobacteria*, *Sphingobacteria*, and *Cytophagia* [72], while *Firmicutes* include 421 genera classified in 7 classes, including *Bacilli*, *Clostridia*, *Erysipelotrichia*, *Limnochordia*, *Negativicutes*, *Thermolithobacteria*, and *Tissierellia* [73]. It is difficult to compare microbiome at phyla level to reach a conclusion about the GM composition; however, it has been observed that children with ASD have a higher count of *Clostridium* (phylum: *Firmicutes*), *Lactobacillus* (phylum: *Firmicutes*), *Sutterella* (phylum: *Proteobacteria*), and *Desulfovibrio* (phylum: *Proteobacteria*), and a lower count of *Bifidobacterium* (phylum: *Actinobacteria*) and *Akkermansia muciniphila* (phylum: *Verrucomicrobia*) [42]. According to a recent systematic review on GM alterations in ASD [74], *Lactobacillus, Bacteroides, Desulfovibrio*, and *Clostridium* were increased in individuals with ASD relative to healthy controls in certain studies, while *Bifidobacterium, Blautia, Dialister, Prevotella*, *Veillonella*, and *Turicibacter* were consistently decreased. The heterogeneity in the results may be attributed to the variations in the methodological approaches used in different studies, such as sample sources, storage temperature, methods of DNA extraction, primers used in PCR, 16S rDNA sequencing platforms, and data analysis methods [74]. Other possible causes of heterogeneity can be related to the study participants’ characteristics, including age, gender, type of control (sibling vs. non-sibling), dietary factors, and GI symptoms [74]. Thus, well-designed studies are needed for a better understanding of the significance of GM in ASD. 

*Desulfovibrio* was shown to be higher in ASD [75,76], and exhibited a positive correlation with the severity of autism manifestations [75]. *Desulfovibrio* is a genus of Gram-negative, sulfate-reducing bacteria and is a major producer of hydrogen sulfide (H2S) in the gut [77]. H2S produced by *Desulfovibrio* was found toxic to the gut epithelial cells of individuals with ASD [78]. High levels of H2S may cause mucus disruption and inflammation, while low levels can directly stabilize the mucus layers of the gut, prevent fragmentation and adherence of the microbiota biofilm to the epithelium, inhibit the invasion of pathogenic bacteria, and help resolve inflammation and tissue injury [79]. Gut *Desulfovibrio* was associated with an increase in the self-reported severity of gastrointestinal symptoms in adults [80], however *Desulfovibrio* abundance was not found to be correlated to GI symptoms in children with ASD [75]. It was noted that there is a dose response in ASD severity with increased *Desulfovibrio*, which was hypothesized to be attributed to its production of lipopolysaccharides (LPS) and reduction of sulfate, otherwise required for other physiological reactions [81]. It has been reported that children with ASD had significantly more ear infections and were given significantly more antibiotics than those who developed normally [78]. *Desulfovibrio* is often resistant to these agents, which suppress certain members of the normal flora, allowing for the overgrowth of *Desulfovibrio* to higher levels, with a higher production of harmful compounds such as LPS and H2S [78]. Because of these toxic byproducts, gastrointestinal inflammation and subsequent neuro-inflammation associated with leaky gut syndrome can occur in children with ASD [82]. 

A surge in bacteria that synthesize propionic-acid such as *Bacteroides* and *Clostridia* (some species) may increase the risk of deterioration in social behavior [42]. Injecting propionic acid into the cerebral ventricles of rats has led to the development of biological, chemical, and pathological changes that are characteristic of autism [83]. In an animal-model study, the effects of prenatal propionic acid on the social behavior of neonatal, adolescent, and adult rats suggested an impairment in social recognition, mediated by the olfactory system in the young rats; adolescent males showed increased levels of locomotor activity compared to females and approached the novel objects more in contrast to the controls [84]. *Clostridium* species counts are consistently high in children with ASD [10,42]. A significant positive correlation was found between the relative abundance of *Clostridium* species and the Childhood Autism Rating Scale score in children with ASD [85]. *Clostridium histolyticum* and *Clostridium perfringens* were consistently reported to be increased in the stool samples of children with ASD [86]. *Clostridium histolyticum* contributes to gut dysfunction [10], and *Clostridia* toxins contribute to behavioral changes by altering neurotransmitter function [87]. Another study reported an increase in *Clostridium histolyticum* in stools from individuals with ASD compared to healthy controls [88]. Reducing *Clostridium* numbers can dampen both gastrointestinal and behavioral symptoms [86]. 

Luna et al. conducted a study on 35 children, 14 of which were diagnosed with both ASD and functional gastrointestinal disorders (FGID), and compared their GM profiles to TD children with and without FGID [49]. Children with ASD and FGID had a different profile than TD children with FGID. Children with ASD had a statistically significant higher number of *Clostridiales*, specifically *Clostridium lituseburense*, *Lachnoclostridium bolteae*, *Lachnoclostridium hathewayi*, and *Flavonifractor plautii*, and a lower number in other *Clostridiales*, including *Dorea formicigenerans* and *Blautia luti* (*p* < 0.05 for all bacteria) [89]. These differences in microbiome between children with ASD also provide an explanation of ASD-related behavior [42]. A difference in the GM has also been observed when challenged with GID. Chronic constipation, for example, can be characterized by a relative decrease in *Lactobacillus* and *Bifidobacterium* [90,91,92], and an increase in potentially pathogenic microorganisms such as *Pseudomonas aeruginosa* and *Campylobacter jejuni* [93]. A decrease in *Veillonella* has been reported in subjects with autism that may disturb the fermentation of lactate in children with ASD [94]. On the other hand, patients with FGID, including irritable bowel syndrome, have higher counts of *Veillonella* and *Lactobacillus*, along with high production of SCFAs including acetic acid and propionic acids [95]. 

In a study comparing GM in ASD children and healthy control groups, the functional analysis of SCFA-producing bacteria demonstrated that butyrate and lactate producers were less abundant in the stools of the ASD group, while mucin-degraders and other SCFA-producers were more abundant in the ASD group [94]. Butyric acid can promote the synthesis of mucin and enhance intestinal tight junction integrity. The depletion of butyrate-producing bacteria and enrichment of mucin-degraders can be linked to the observed abnormal intestinal permeability reported in individuals with autism [96]. Butyrate is recognized as an anti-inflammatory SCFA that contributes to colon health and has been demonstrated to rescue ASD cells during oxidative stress and enhance mitochondrial function in the context of physiological stress. Butyrate can also modulate the synthesis of the neurotransmitters dopamine, norepinephrine, and epinephrine. Butyrate can also modulate neurotransmitter gene expression and the ASD-related genes in cell line models [97]. 

In children with ASD, differences in GM profiles reported some common species that were correlated with GID. Table 1 lists the types of bacteria that were found to be significantly altered (increased or decreased) in children with ASD suffering from GID. 

Additionally, variation in GM between children with ASD and TD children, both presenting with GID, were noted. For example, children with ASD had decreased levels of *Flavonifractor plautii, Bacteroides eggerthii, Bacteroides uniformis, Flavonifractor prausnitzii,* and *Clostridium clariflavum* in functional constipation [89], an increase in *Clostridium aldenense*, decreases in *Bacteroides luti, Bifidobacterium adolescentis, Eubacterium ventriosum, Anoxystipes fissicatena, Coprococcus comes, Eubacterium ramulus,* and *Phascolarctobacterium faecium* in aerophagia [89], decreases in *Akkermansia muciniphila, Coprococcus catus, Odoribacter splanchnicus, Clostridium lactatifermentans*, and *Ruminococcus lactaris* in abdominal migraine [89], an increase in *Clostridium aldenense* in irritable bowel syndrome [89], and increases in *Bacteriodaceae, Lachnospiraceae, Prevotellaceae*, and *Ruminococcaceae* in abdominal pain [106]. 

Alpha diversity is defined as the mean diversity of species (or other taxonomic groups) within a microbial community. Alpha diversity metrics summarize the structure of a community with respect to its richness (number of taxonomic groups), evenness (distribution of abundances of the groups), or both [107]. Based on a recent systematic review, there were no consistent patterns of alpha diversity in children with ASD compared to siblings and healthy controls, though a few studies reported significant differences [108]. A study comparing the GM in ASD and healthy children in China reported a significant increase in bacterial richness in the ASD group compared with healthy children [109]. Another study compared the GM of subjects with autism with gastrointestinal symptoms, siblings not showing autistic symptoms (sibling controls), and the non-sibling control subjects. Significantly more diversity and richness of microbial communities were found in ASD compared to the other groups [76]. The increased richness in ASD children was attributed to an increase in the harmful genera or species contributing to the severity of autistic symptoms (listed in Table 1). 

The analysis of the beta diversity (a measure of diversity of microbial community composition between different groups) showed that the GM in the ASD group was distinct from that of the healthy control group [109]. However, there is inconsistency in the available data regarding significant differences in the beta diversity between ASD and siblings without autistic symptoms and non-sibling control subjects. For instance, a study investigating GM in ASD children with and without GI dysfunction reported that there was no clustering of samples from ASD children, regardless of the autism severity and GI dysfunction. This indicated a non-significant difference in the bacterial composition within and between the autistic and control groups [110]. The same study also indicated that there were non-significant differences within the ASD group when comparing those with and without GI dysfunction.

GM composition is influenced by a variety of factors, including age, body mass index, and genetic disposition, and other external factors such as diet, antibiotic treatments, exercise frequency, and environmental pollutants [38,111,112,113]. Dietary factors influence GM through acting as substrates to different microbial species, and through altering the gut environment (e.g., pH levels and bile acid (BA) concentrations), influencing microbial growth [111]. Dietary approaches with varying compositions of fiber, proteins, and fats, such as the Western diet, Mediterranean diet, and others, promote the growth of different species of bacteria [114]. Of these macronutrients, both the quantity and quality of dietary fats are proposed to be major factors in promoting this change due to their direct effects on BA concentrations [111]. These associations will be explored more closely in Section 3.

Dietary interventions are poised to play a key role in the brain-gut-microbiome axis in order to maintain both the brain health and gut health by preserving a healthy balance of the GM. In children with ASD, who already have an altered GM profile, translating the effects of external factors such as diet into clinical practice can lead to the potential treatment or management of ASD symptom severity and GID. 

## 4. Dietary Fat and Gut Health

The term fat usually refers to all “fatty” substances found in food and the body [115]. In dietary terms, “lipids” is a broader term which encompasses fats, usually in the form of triglycerides (TG), cholesterol, and phospholipids, all of which are found in varying concentrations in different foods [115]. TG make up about 95% of all lipids in the diet [116], and are composed of one glycerol molecule and three attached fatty acids. The fatty acids are mainly what define the health and metabolic effect of the fat. They can either be saturated, monounsaturated, or polyunsaturated fatty acids based on the number of double bonds that exist in the carbon–carbon skeleton of the fatty acid [115]. 

Theoretically, fatty acids are not thought of as a diet source for gut bacteria, due to the need of oxygen for the metabolism of fat, and the gut is inhabited mostly by anaerobes [117]. Nonetheless, bacteria has been found to interact with fat, as fatty acid intermediates have been detected in the intestine [117]. Some fatty acids exert antimicrobial activity that affects the survival of certain bacterial species [117,118]. For instance, bacterial growth hindrance or cell death may result from the toxic compounds generated by the action of free fatty acids (FFAs) on the cell membrane, causing disruption of enzymatic activities, leading to interferences with energy production, and impairment in nutrient absorption [43]. The presence of fat in the distal intestines may induce a loss of diversity in the microbiome community due to this antimicrobial activity [119]. Due to this effect of fat on the microbiome, the composition of fat, and its effects on the GM, are interesting areas of investigation; being easily manipulated through diet can be a target for non-invasive preventive and therapeutic interventions.

### 4.1. Saturated Fatty Acids

Saturated fatty acids (SFAs) are composed only of carbon –carbon single bonds, and are usually solid at room temperature [115]. While they are primarily found in animal products, such as meat, butter, and dairy products [120], they are also found in large concentrations in tropical plant oils, such as palm and coconut oils [115]. 

High SFAs intake has been generally reported to have a negative influence on GM, which could increase correlated GID, as illustrated in Figure 2. One of the major mechanisms includes an increase in BA concentration [121], which has a direct antimicrobial effect on GM [122]. BAs are secreted as a result of fat ingestion, specifically as a response to cholecystokinin [123]. Approximately 94% of BAs are absorbed from the distal small intestines, and the rest are lost through feces. Primary BAs (e.g., cholic acid and chenodeoxyvholic acid) received from the liver can be converted into secondary BAs (deoxycholic acid and lithocholic acid [124]) by gut bacteria [119] [123,125]. BAs play an interesting role in gut health, as they have a potential pathogenic role in inflammation on one hand, and in intestinal mucosal defense on the other [126]. For example, ursodeoxycholic acid administration resulted in favorable intestinal changes, reduced intestinal permeability, and decreased free radical generation; however, at higher doses, the same BA was found to be cytotoxic [126]. Preserving the dynamic equilibrium between GM and BA composition is essential to avoid pathological states [122]. Secondary BAs were found to be accumulated in individuals following the Western diet, especially deoxycholic acid [122]. These secondary BAs are more potent antimicrobial agents [122], and have been associated with gut inflammation by causing oxidative stress, cytotoxicity by causing damages to the cell membrane, and can even be carcinogenic [119,125,127]. To demonstrate the effect of a high input of BAs, feeding rats high levels of cholic acid led to an expansion of the secondary BA-producing bacteria (of the phylum *Firmicutes*), along with an increase in *Firmicutes* from 54% in the control group to 93–98%, including a significant expansion in many species found in *Clostridium* cluster XIVa [122]. This increase in BAs also led to a reduction of *Bacteroidetes* and *Actinobacteria* [122]. The overlap of GI and ASD symptoms may be explained by shared pathophysiological mechanisms that include intestinal inflammation, as evident from the endoscopy in children, with ASD showing that different segments of the GI tract may be affected with inflammation, as upper and lower endoscopy results showed reflux esophagitis, chronic gastritis and duodenitis, and ileocolotis [128]. Although ASD is a genetically determined brain disorder, GI inflammation may contribute to brain dysfunction [128]. 

Interestingly, studies have found that SFAs can change the BA pool and thus contribute to intestinal inflammation and alter the microbiome [119], specifically in regards to the bacterium *Bilophila wadsworthia*, including increased inflammation, increased intestinal permeability with higher mucosal, and systemic immune responses [129]. These effects may increase the likelihood of abdominal pain and GID, such as diarrhea [130]. Milk-fat (high in SFA) consumption also promotes the growth of *Bilophila*
*wadsworthia* and intestinal inflammation, as this sulphate-reducing bacteria can increase the concentration of hydrogen sulphide and disrupt the epithelial tissue in the intestinal mucosa [131]. SFA can also contribute to GID due to other microbiome-related changes, including the increase in *Clostridium* clusters. de Wit et al. (2012) conducted a trial on mice that received four diets, either low-fat, high-fat with high SFA (palm oil), high-fat with high MUFA (olive oil), or high-fat with high polyunsaturated fatty acids (PUFA) (safflower oil) After 8 weeks, the group that consumed palm oil (high SFA) had an increased *Firmicutes*/*Bacteroidetes* ratio. *Firmicutes* had a high proportion of *Clostridium* clusters XI, XVII, and XVIII. *Clostridium* cluster XI specifically contributes to the production of deoxycholic acid and is considered a harmful bacteria for the large intestines [132]. Moreover, high palm oil intake was associated with reduced microbial diversity [132]. 

Microbial diversity is an important marker of health [129], as the diversity of bacterial species maintains the intestinal ecosystem [133], increases the functional resilience of microbial community [117], and decreases shifts in the microbiome as a response to environmental changes (e.g., diet) [129]. For instance, patients with inflammatory bowel disease exhibit lower microbial diversity that increases during remission [133]. High SFA intake through palm oil intake also induces an overflow of fatty acids to the distal parts of the intestines [132]. The concentration of GM increases steadily along the gastrointestinal tract, starting from 10^1^ bacteria/gram content in the stomach, to 10^12^ bacteria/gram in the colon [134], and the bacteria inhabiting the stomach and proximal duodenum are usually bile resistant [134]. An overflow of dietary fats to the distal intestines is suggested to be strongly involved in changing the GM composition [135], also observed after administering a high palm oil diet [132]. This overflow of dietary fats from palm oil was greater than safflower or olive oils [132].

A review by Nogay and Nahikian-Nelms (2021) also supported the increase in the *Firmicutes* in mice. Observational studies comparing the diets of children following either mainly the Western diet or a predominantly vegetarian diet reported profound differences in the distribution of bacteria phyla, in which the Western diet showed an increased prevalence of *Firmicutes*, and a decreased prevalence of *Bacteroidetes* and *Actinobacteria* [136]. High SFA intake has been suggested to have promoted the growth of pathogenic bacteria through increasing taurine-conjugated BAs [137]. A high intake of SFA through animal-based protein, one of the hallmarks of the Western diet [138], increased the number of *Clostridia* and decreased the count of *Bifidobacterium adolescentis* [139]. *Bifidobacterium adolescentis* was reported to downregulate nuclear-factor kappa B [140], a protein complex that induces an inflammatory state [141] and inhibits the production of LPS in the gut [140]. LPS are endotoxins found in Gram-negative bacteria, much of which are part of the human gut. *Bacteroides*, for example [142], have been found to actually increase with high-fat feeding [143], along with *Clostridium* and *Ruminoccoccus* [42]. LPS have been found to increase visceral hypersensitivity [144], which is implicated in the pathogenesis of abdominal pain and discomfort [145]. 

Bile-tolerant anaerobes such as *Bilophila wadsworthia* and the *Alistipes* genus increase in the Western diet along with high milk-fat consumption [146]. The *Alistipes* genus is relatively new, and is reported to have contrasting evidence regarding its pathogenic and/or protective roles [147]. A study which analyzed the microbiome of 22 children with irritable bowel syndrome found an abundance of the *Alistipes* genus (four specific taxa within this genus) and a positive correlation with a greater frequency of pain [148]. 

The reviewed literature suggests that high intakes of SFA exert unfavorable effects on the GM composition, leading to the overgrowth of specific species and reduced GM diversity that have been correlated with GID. Dietary patterns with high SFA levels such as the Western diet should be discouraged. Furthermore, these GM changes can translate into other metabolic disorders such as insulin resistance and inflammation [114].

### 4.2. Unsaturated Fatty Acids

Unsaturated fatty acids (UFAs) contain one or more double bonds in the carbon–carbon skeleton of the fatty acids, are predominantly found in plant products, and are usually liquid at room temperature [115]. Those with only one double bond, referred to as monounsaturated fatty acids (MUFA), are found in highest concentration in olive, peanut, and canola oils, avocado, and some nuts and seeds; those with two or more double bonds are referred to as polyunsaturated fatty acids (PUFA), and are found in other oils (sunflower, corn, and soybean), walnuts, flaxseeds, and fish [115]. 

Essential fatty acids (EFA) are a category of PUFA that are not synthesized by humans, and include both omega-3 (alpha-linolenic acid) and omega-6 (linoleic acid) and their derivatives, and are thought to exert a positive effect on GI health through their effect on the GM (Figure 2) [149]. Children with ASD have been found to have disturbed concentrations of omega-3 and omega-6, and decreased omega-3 levels [87,119,150], possibly due to nutritional deficiency or genetic defects [87]. An adequate balance of EFA is essential for brain development, and is therefore considered a possible biomarker for ASD, and omega-3 supplementation can potentially improve behavioral measures in children with ASD [151]. Low omega-3 levels, or a disturbed omega-3/omega-6 ratio, increases inflammatory cytokine and oxidative stress [150]. Dietary flaxseed oil (high omega-3) was found to significantly reduce proinflammatory biomarkers such as plasma IL-6, IL-1b, and TNF-a, in rats [149]. GID in which inflammation plays a role may benefit from Docosahexaenoic acid (DHA), an omega-3 supplementation [152]. Menni et al. (2017) noted that increased serum DHA was associated with bacterial groups that were negatively correlated with intestinal inflammation and Crohn’s disease [153]. In addition, low omega-3 levels can lead to the malfunction of neurotransmitters, as EFA can modulate the expression and action of neurotransmitters including serotonin, dopamine, and acetylcholine [154].

Watson et al. reported that, following an 8-week supplementation with omega-3 in healthy middle-aged women, *Bifidobacterium*, *Roseburia*, and *Lactobacillus* increased [155]. *Roseburia* is considered to play a major role in gut health due to its anti-inflammatory and immunity-maintenance properties [156]. In one study comparing children with ASD to healthy children, some *Roseburia* species were higher in children with ASD and others were higher in healthy children [157]. Menni et al. (2017) also reported the strongest correlation of omega-3 with an increase in the *Lachnospiraceae* family, which includes *Roseburia*. [153]. Omega-3 administration in rats also showed a favorable impact on GM. Comparing different types of fats, corn oil diets increased the abundance of *Firmicutes* in rats, which was reversed through the administration of flaxseed oil. Therefore, flaxseed oil reduced the *Firmicutes*/*Bacteroidetes* ratio and also reduced *Blautia* abundance [149]. Furthermore, the abundance of *Firmicutes* and *Blautia* was significantly correlated with higher LPS, while *Bacteroidetes* was correlated with lower LPS levels [149]. Although the theoretical evidence supports the possibility of favorable role of omega-3 on GM, a recent observational study on 120 children with ASD judged that there were actually no significant associations between GI symptoms and dietary omega-3 intake [158]. Additionally, in a supplementation trial of omega-3 in children with ASD, there were no significant differences in the occurrence of adverse events, including a range of GI symptoms in omega-3 supplemented and placebo groups [159]. In 19 patients who received the omega-3 supplement, there were complaints of abdominal pain (*n* = 6), diarrhea (*n* = 5), and constipation (*n* = 7) [159]. Considering the high prevalence of GID in ASD, a treatment that does not cause GID warrants high importance. 

Regarding high UFA diets, fish oil-fed and lard-fed mice were compared; an increase in *Actinobacteria* (including *Bifidobacterium*), lactic acid bacteria (including *Lactobacillus*), and *Verrucomicrobia* (including *Akkermansia muciniphila*) was observed in the high fish oil diet [160]. In contrast, lard-fed mice showed an increase in *Bacteroides*, *Turicibacter*, and *Bilophila* [160]. On the other hand, a decrease in *Bifidobacterium* after the addition of walnuts and almonds (high in MUFA and PUFA), and an increase in other genera, including *Clostridium* and *Roseburia*, were observed [161,162]. *Akkermansia muciniphila* was significantly decreased in children with ASD and was associated with abdominal migraine [89]. 

The data on MUFA and GM is limited and inconsistent, therefore its effect is still inconclusive. Singh et al. (2017) reviewed that MUFA does not lead to significant shifts in any bacterial genera. Oleic acid, a type of MUFA found in olive oil, increases *F. prausnitzii* [163], contradictory to a previously reported decrease in cases of constipation [89]. *F. prausnitzii* is a positive indicator of intestinal health due to its anti-inflammatory properties [163]. In contrast, Bailey and Hoschler (2018) found that, when mice were either fed canola oil (high MUFA) or palm oil (high SFA), the canola oil group had a higher plasma LPS. Supplementation with omega-3 fatty acids and consumption of high oleic canola oil have shown to increase the levels *Faecalibacterium* belonging to the *Firmicutes* phyla [164]. Mandal et al. (2016) concluded that MUFA intake in pregnant women was associated with an increase in *Firmicutes*, *Proteobacteria* and *Bacteroidetes*, and a reduction of *Actinobacteria*/*Proteobacteria* [165]. An increase in *Proteobacteria* was similar to SFA, as it is pathogenic and proinflammatory [165]. Due to the conflicting evidence on MUFA, and the lack of literature in children specifically with ASD, the effect remains inconclusive. 

UFA is suggested to have a beneficial impact on GM and gut health. While it could be postulated that replacing SFA with UFA would lead to a more favorable impact on GM, mainly because of the beneficial effect in the limited availability of SFAs, the results are inconsistent and inconclusive. Omega-3 is hypothesized to play a positive effect on GM; still, experimental evidence to support this association, specifically in children with ASD, is lacking. 

### 4.3. High-Fat Diets

Various dietary interventions are proposed for children with ASD, the ketogenic diet is one of them [166]. The ketogenic diet is a dietary pattern that is high-fat, moderate-protein, and low-carbohydrate, used traditionally for controlling seizures in patients with epilepsy [167]. A systematic review summarized that such a diet was promising in attenuating ASD-related behaviors in mice and humans [168]. However, ketogenic dieting should be recommended with caution, as the high-fat percentage (65–90%) can be associated with GID and may lead to nutritional deficiencies, due to low tolerability and the high food selectivity in children with ASD [150]. Gastrointestinal side effects of ketogenic diets in children with ASD included constipation, diarrhea, and vomiting, which occurred within 2–4 weeks of the ketogenic diet onset [169]. 

Due to the need of BAs for the emulsification of dietary fat, an increase in dietary fat intake is correlated with an increase in the production and secretion of BAs, which are consistent with a higher level of secondary BA in fecal samples [124]. Thus, the consumption of high-fat diets encourages the growth of bile-tolerant bacteria, which are usually Gram-negative bacterial species [119]. Although bacteria from the class *Clostridia* are Gram-positive, they are also bile-resistant, and therefore increase with high-fat feeding. These include the two families of *Ruminococcaceae* and *Blautia* [119]. These families may be negatively associated with gut health, and increased abdominal symptoms in animal models [170]. Regarding human studies, in a Swedish study of 159 individuals (52 cases of abdominal pain and 107 controls with no pain), *Blautia* was significantly increased in the cases with abdominal pain (*p* = 0.045) [170]. Mimicking the effect of high-fat diets by providing cholic acid (BA), supplements resulted in an increased *Firmicutes*/*Bacteroidetes* ratio and decreased bacterial diversity [119]. More specifically, in an animal model study, the authors concluded that the increase in *Firmicutes* was driven by an increase in *Clostridia* and *Erysipelotrichi* classes [171]. From *Clostridia*, the genus *Blautia*, and the genus *Allobaculum* from *Erysipelotrichi*, dominated [171]. The results on *Allobaculum* were inconsistent, as another animal study found that its level decreased in high-fat diets, but increased with high-protein diets [172]. This data indicates that high-fat diets may have unfavorable effects on the GM profile, and a higher risk of GID, due to the association of dysbiosis with gut health. Other effects of high fat diets include decreasing the *Roseburia* species [173]. 

Another concern with high-fat diets is related to the increase in LPS. Dietary fat influences GM through: (1) an increase in the absorption of LPS, related to chylomicron formation, and (2) the growth of Gram-negative bacteria, which increases LPS concentrations [142]. LPS can migrate, along with the chylomicron, from the intestines to systemic circulation through absorption [142], with a potent inflammatory effect that contributes to the development of metabolic diseases [174]. In the intestines, LPS may induce severe inflammation and increases the risk of gut-related complications, including diarrhea and abdominal pain [175]. A diet consisting of 72% energy from fat resulted in a 2–3 times increase in circulating LPS in a murine animal study [119]. Additionally, increased LPS is a significant marker of consideration, independently and inversely correlated with lower socialization scores in subjects with ASD [176]. In contrast, *Bifidobacterium* increases as a response to low-fat diets [177], and plays a beneficial role in decreasing LPS levels [178], contributing to restoring intestinal health.

Mediated through the effects of high concentrations of BAs and LPS, it is possible that high-fat diets increase the risk of GID, especially in children with ASD for whom ketogenic diets are considered. Further research with long-term trials is warranted to develop appropriate dietary fat recommendations, particularly in the context of children with ASD.

## 5. Limitations and Recommendations

There are some limitations in this review. Most of the evidence presented is extracted from either animal studies or from human studies that present several limitations, such as the inclusion of observational study designs and small sample sizes, thus introducing a risk of bias. Lang et al. (2018) concluded that demographic and anthropometric traits overpower the effect of diet in influencing GM. Heterogeneity in the results regarding GM composition may also be due to methodological variations [74]. Therefore, robust study designs with large samples can help in controlling the confounding factors and present more conclusive evidence. The lack of studies exploring dietary fat specifically in children with ASD is the major limitation. Nevertheless, this review highlights a gap in the literature, creating the need for future studies utilizing an experimental approach to explore the effect of dietary interventions with varying quantities of total fat, fatty acid composition in GM, and the presence of GID, as it can be employed as a promising therapeutic approach. Furthermore, the relation between the severity of behavioral ASD symptoms with GM composition and GID must be explored in future studies. This will help in designing interventional protocols through appropriate dietary regimes, particularly in regard to dietary fats, to reduce the severity of the disease in children with ASD. 

## 6. Conclusions

Data from the literature suggests that children with ASD have different microbiome profiles than TD children, which contribute to both behavioral disorders and GID. Some of these bacteria are increased or decreased by modulating dietary fat. This review proposes that dietary fat intake, as part of the overall diet, may influence the presence and severity of GID through various mechanisms, although experimental data is too scarce to reach affirmations. SFAs were consistently shown to be associated with a negative impact on GM, which could be translated into GID based on the available evidence. Evidence regarding UFAs is inconclusive due to the inconsistent reporting of results; however, omega-3 is hypothesized to reduce GID due to its anti-inflammatory effect and its favorable impact on the GM. High-fat diets exhibited a negative influence on GM, mainly through increased LPS and BA levels, however data is insufficient to suggest specific recommendations. As GID are related to gut dysbiosis, dietary interventions based on GM modulation can serve as an effective therapeutic tool to treat the GID-associated symptoms and severity in children with ASD by restoring the normal GM, limiting inflammation, and restoring epithelial barrier function. 

## Figures and Tables

**Figure 1 nutrients-13-03818-f001:**
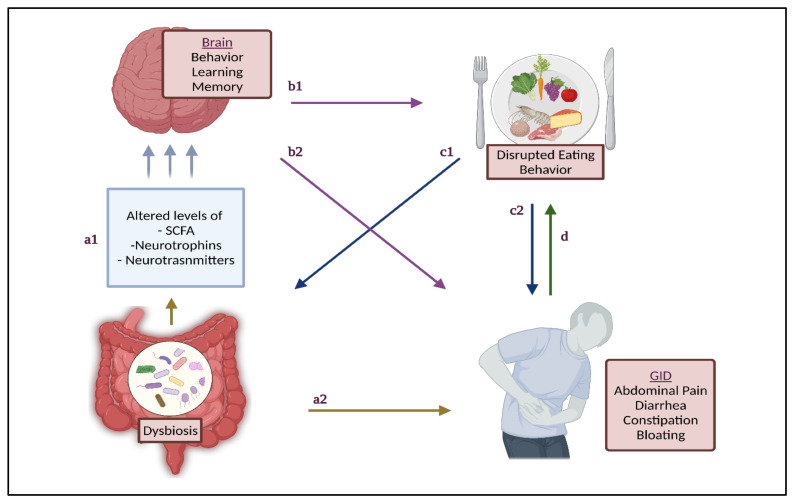
The brain–gut–microbiome axis in the etiology of gastrointestinal disorders in autism spectrum disorder: (**a1**) Signaling molecules from GM communicate with various organs, including the brain affecting ASD-related behavior. (**a2**) Dysbiosis contributes to GID. (**b1**) ASD severity may increase risk of disrupted eating behavior, affecting the quantity and quality of food intake. (**b2**) ASD severity manifests as a lack of ability to communicate GID presence, contributing to increased symptoms and severity. (**c1**) Disrupted eating behavior compromises diet quality, contributing to altered GM composition. (**c2**) Disrupted eating behavior contributes to GID directly and indirectly through (**a2**). (**d**) GID presence and severity contributes to compromised eating, thereby disrupted eating behavior. ASD: autism spectrum disorder; GID: gastrointestinal disorders; GM: gut microbiome; SCFA: short-chain fatty acids.

**Figure 2 nutrients-13-03818-f002:**
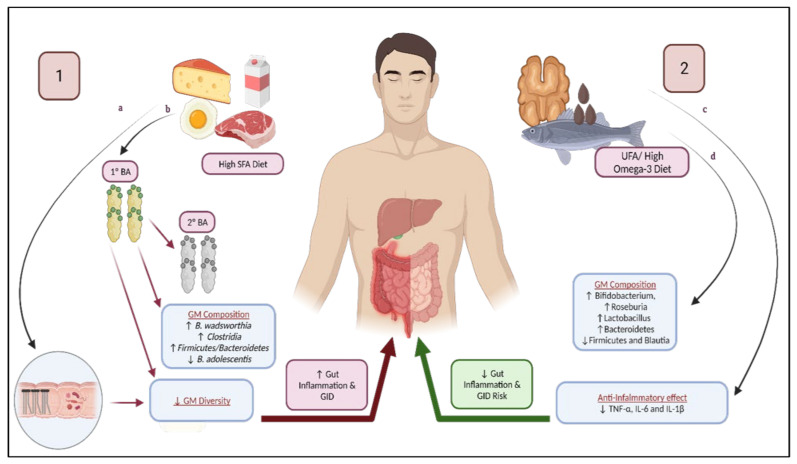
Dietary fat composition and risk of gastrointestinal disorders: (**1**) High SFA diet contributes to GID through modulation of GM. (**a**) SFA increases FA overflow to the distal intestines, reducing GM diversity, thus contributing to GID. (**b**) SFA increases the levels of primary and secondary bile acids, contributes to changes in GM composition, reduced GM diversity, and gastrointestinal inflammation, thus increasing the risk of GID. (**2**) Unsaturated fats, particularly high omega-3 diets, reduces GID risk due to (**c**) significant reduction of proinflammatory cytokines and (**d**) its favorable impact on GM composition and diversity, increasing SCFA levels. BA: bile acid; GM: gut microbiome; GID: gastrointestinal disorders; SFA: saturated fatty acids; SCFA: short-chain fatty acids.

**Table 1 nutrients-13-03818-t001:** Variations in bacterial species with functional gastrointestinal disorders in children with autism spectrum disorder.

GID	Disorder Definition	Increase in:	Decrease in:	Ref.
Functional Constipation	Hard, infrequent bowel movements without an organic etiology [98]	*Escherichia/Shigella* and *Clostridium cluster XVIII*	--	[99]
		*Fusobacterium, Barnesiella, Coprobacter and Actinomycetaceae*	*Butyrate-producing taxa*	[100]
		*--*	*Turicibacter, Roseburia, Dialister, Staphylococcus, Butyricicoccus, * *Faecalibacterium, Gemmiger*	[101]
		*Lachnospiraceae NK4A136, Subdoligranulum, Ruminococcus, Barnesiella, Butyricicoccus, and Ruminiclostridium*	*Fusobacterium, Acidaminococcus, and Veillonella*	[102]
Abdominal Pain	Abdominal pain accompanying various GID	*Turicibacter sanguinis, Clostridium aldenense, Clostridium lituseburense, Flavonifractor plautii, Clostridium disporicum, Clostridium tertium, Tyzzerella species and Parasutterella excrementihominis*	*--*	[89]
		*Ruminococcus torques*	*--*	[103]
		*Bacteroides fragilis*	*--*	[104]
		*Dorea, Prevotella*	*Bacteroides, Roseburia*	[105]

## Data Availability

Not applicable.

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
