# Peer review of "Dietary Fat Effect on the Gut Microbiome, and Its Role in the Modulation of Gastrointestinal Disorders in Children with Autism Spectrum Disorder"

_nutrients, 2021, doi:10.3390/nu13113818_

Round 1

Reviewer 1 Report

The review is very well written and nicely organized. 

The authors can comment how age and sex might play a role in GI disorders in autism.

Also  can GI dysregulation be predicted early in autism?

Author Response

#

Comment

Response

Line #

1

The review is very well written and nicely organized.

Thank you for the encouragement

--

2

The authors can comment how age and sex might play a role in GI disorders in autism

The role of age and sex was added in the introduction

69-72

3

Also, can GI dysregulation be predicted early in autism?

Early onset of GI dysregulation in autism was added to the introduction

67-69

Reviewer 2 Report

The authors provided a thorough review on the role and effect of gut microbiome on gastrointestinal disorders in children with ASD, which is a topic of outstanding. 

Authors provided a comprehensive and quite well organized review.

However, I would suggest to provide a better methodological specification: didi the authors aimed to provide a narrative review on the topic? or is it a scoping/systematic review? In the latter case, it would be advisable to specify the procedures used to collate data (PRISMA guidelines). Moreover, from the methodological point of view, it would be advisabele to better specify inclusion and exclusion criteria of the studies. 

In my opinion it mught be not necessary to provide statistics from cited studies (e.g. line 169-170). 

Finally, a scoping review on efficacy and feasibility of Ketogenic dietary treatments in pediatric patients has been recently conducted (Varesio and colleagues, 2021), it might be indicated to cite it in the 3.3. paragraph. 

Author Response

#

Comment

Response

Line #

1

The authors provided a thorough review on the role and effect of gut microbiome on gastrointestinal disorders in children with ASD, which is a topic of outstanding.

Thank you for the encouragement

--

2

Authors provided a comprehensive and quite well-organized review.

Thank you for the encouragement

--

3

However, I would suggest to provide a better methodological specification: did the authors aimed to provide a narrative review on the topic? or is it a scoping/systematic review? In the latter case, it would be advisable to specify the procedures used to collate data (PRISMA guidelines). Moreover, from the methodological point of view, it would be advisable to better specify inclusion and exclusion criteria of the studies. 

This is a narrative review, not a systematic/scoping review. A paragraph providing an overview of the methods is added. 

140-155

4

In my opinion it might not be necessary to provide statistics from cited studies (e.g. line 169-170).

Statistics were removed and sentence was rephrased

253-255

5

Finally, a scoping review on efficacy and feasibility of Ketogenic dietary treatments in pediatric patients has been recently conducted (Varesio and colleagues, 2021), it might be indicated to cite it in the 3.3. paragraph. 

Thank you for suggesting this important reference. Data were added in section 4.3 (previously 3.3) 

599-601

Reviewer 3 Report

Gut microbiome, especially its role in various disease and disorders is a hot topic at the moment. There are established link between gut bacteria and gut condition such as constipation/IBD/IBS. Some studies came out in recent years suggest a link between gut bacteria and the brain.

This review aims to check existing evidences of gut bacteria and ASD. Furthermore, how dietary fat affects bacteria, and how this can help modulate ASD.

We can see the urgent need of better understanding of ASD. Parts about microbiome is weak, especially most of microbiome reference come from other reviews of ASD and gut microbiome, for instance, reference 16, 17, 18, 24, 35, 36. There are some old references such as 21. Ways to recognise bacteria have revolutionised from culturing, PCR, FISH, to 16s sequencing, or even metagenomic sequencing. Quality of data is important in determine microbiome composition. Introduction is alright, and could do with better microbiome part. GM and ASD part is weak in connecting GM with ASD, especially I see references contradicting each other. If connection between GM and ASD cannot be established, the next part using diet to modify GM therefore improve ASD won’t stand. Because certain bacteria increase and decrease in ASD in different studies; how dietary fat change these bacteria/help ASD is not convincing. There are a few parts mentioned dietary fat and gut inflammation. It seems significant, but I don’t see earlier discussion of ASD with gut inflammation. Besides, if you want to modulate ASD by changing gut bacteria, why diet fat? Is there an issue with ASD dietary fat? How significant is this high fat diet in ASD? If not, why dietary fat not probiotics/prebiotics which are more efficient in changing gut bacteria?

Another thing is many arguments are brought by literature review reference. Should check research data to support your arguments, especially existing literature review can interpret research not in the way they were from the beginning. The last thing is authors need more knowledge of gut bacteria/microbiome and their metabolism, there were a few wrong statements in the manuscript. A few mis-spelling of bacteria names such as desulfovibrio and C.histolyticum.

  1. Abstract line 16 GM plays essential role in nervous development and brain maintenance. It’s essential for immune development, can you put some solid evidence about nervous and brain?
  2. Line 186-192, why is bacteroidete not italic? Bacteroidetes are the major propionate producer in the gut which decreased in ratio, but then in the animal study they also contribute to the risk of ADS. Besides, not all clostridia produce propionate. Desulfovibrio (spelled wrong in your manuscript) produce H2S not propionate.
  3. Clostridia is a big genus and huge differences within this genus. Hard to generalise they are all bad.
  4. Line 222 acid producing bacteria? Pretty much all anaerobic bacteria fermentation ends up with acid production.
  5. Table 1 has some contradicting pairs. Reference 63 had increase in Escherichia/shigella, but 64 said Enterobacteriaceae decreased. Turicibacter in 64 and 54. Bacteroidetes in 63 and 70. By the way, in 63, Bacteroidetes don’t produce butyrate.
  6. Line 318 diversity is important, anything about ASD and bacteria diversity?
  7. Line 325 101 bacteria/gram? 1012?

Author Response

#

Comment

Response

Line #

Gut microbiome, especially its role in various disease and disorders is a hot topic at the moment. There are established link between gut bacteria and gut condition such as constipation/IBD/IBS. Some studies came out in recent years suggest a link between gut bacteria and the brain. This review aims to check existing evidences of gut bacteria and ASD. Furthermore, how dietary fat affects bacteria, and how this can help modulate ASD.

1

We can see the urgent need of better understanding of ASD. Parts about microbiome is weak, especially most of microbiome reference come from other reviews of ASD and gut microbiome, for instance, reference 16, 17, 18, 24, 35, 36. There are some old references such as 21.

1.   Discussion of the microbiome was improved (see response to comment #4)

173-176, 180-184, 190-196,

207-210*, 211-218, 219-229, 273-283, 313-318, 332-334,  

344-354

2.  Reference 21 was deleted where it was mentioned by itself. However, kept when it’s mentioned along another reference (Now Ref. 33)

172, 177, 382

3.   Regarding other references, some references were changed, others supported with more references

284-296,

482-487,  487-489, 497-498, 563-570,

619-621, 630-632, 632-633, 633

2

Ways to recognize bacteria have revolutionized from culturing, PCR, FISH, to 16s sequencing, or even metagenomic sequencing. Quality of data is important in determine microbiome composition.

Improved by adding more information

157-168

3

Introduction is alright, and could do with better microbiome part.

Improved by adding more information

81-89, 92-110, 111-119, 128-130*

4

GM and ASD part is weak in connecting GM with ASD, especially I see references contradicting each other. If connection between GM and ASD cannot be established, the next part using diet to modify GM therefore improve ASD won’t stand.

Improved by adding more information

173-176, 180-184, 190-196,

207-210*, 211-218, 219-229, 273-283, 313-318, 332-334,  

344-354

5

Because certain bacteria increase and decrease in ASD in different studies; how dietary fat change these bacteria/help ASD is not convincing. There are a few parts mentioned dietary fat and gut inflammation. It seems significant, but I don’t see earlier discussion of ASD with gut inflammation.

Additional discussion about gut inflammation in ASD has been added to support that association.

The role of dietary fat is proposed due to its documented role on various bacteria species, which is identified as a gap in individuals with ASD. This is highlighted in the limitations section, and recommendations for this area of research have been suggested.

444-450

6

Besides, if you want to modulate ASD by changing gut bacteria, why diet fat? Is there an issue with ASD dietary fat? How significant is this high fat diet in ASD? If not, why dietary fat not probiotics/prebiotics which are more efficient in changing gut bacteria?

Prebiotics/Probiotics therapy is already a promising approach that we have acknowledged in the Introduction. However, differences have been observed in food preferences and dietary fat intakes between children with ASD & TD children. Additionally, it can impact GM composition. Therefore, this review aimed to explore the role dietary fats in ASD and GID, an area that needs due attention

116-119

7

Another thing is many arguments are brought by literature review reference. Should check research data to support your arguments, especially existing literature review can interpret research not in the way they were from the beginning.

Some of these references were updated as mentioned in the response 3 of comment #1

284-296,

482-487,  487-489, 497-498, 563-570,

619-621, 630-632, 632-633, 633

8

The last thing is authors need more knowledge of gut bacteria/microbiome and their metabolism, there were a few wrong statements in the manuscript. A few mis-spelling of bacteria names such as desulfovibrio and C.histolyticum.

Fixed mis-spellings

298, 281

9

Abstract line 16 GM plays essential role in nervous development and brain maintenance. It’s essential for immune development, can you put some solid evidence about nervous and brain?

Statement was re-phrased in the abstract. Additional references on GM and brain were added in the main text

16-18, 81-89, 92-110, 211-218, 219-229

10

Line 186-192, why is bacteroidete not italic? Bacteroidetes are the major propionate producer in the gut which decreased in ratio, but then in the animal study they also contribute to the risk of ADS. Besides, not all clostridia produce propionate. Desulfovibrio (spelled wrong in your manuscript) produce H2S not propionate.

Italic & bacteria name corrected

485, 281

Bacteroides (of the phylum Bacteroidetes) are increased in ASD, which are propionate producers. In addition, Clostridia contribute to neurological changes, some of them are also propionate producers. The discussion of the ratio was removed from that paragraph as it is mentioned in Lines 272-273

288-289

11

Clostridia is a big genus and huge differences within this genus. Hard to generalize they are all bad.

We have specified some of the species within Clostridia that dampen both gastrointestinal and behavioral symptoms in ASD children  

296-300, 307-310, Table 1

12

Line 222 acid producing bacteria? Pretty much all anaerobic bacteria fermentation ends up with acid production.

This statement was changed and mentioned specific types of bacteria.

313-316

13

Table 1 has some contradicting pairs. Reference 63 had increase in Escherichia/shigella, but 64 said Enterobacteriaceae decreased. Turicibacter in 64 and 54. Bacteroidetes in 63 and 70. By the way, in 63, Bacteroidetes don’t produce butyrate.

The Table has been revised:

1.     The decrease in Enterobacteriaceae was incorrectly reported from ref 64. It was deleted

2.     Regarding Turicibacter, these were mentioned as reported by each article. No further analysis or interpretation were made based on these results.

3.     References that compared between children with ASD and TD children were removed

4.     Reference 63- results in the article stated that it is butyrate-producing taxa, without providing information on specific species. However, Bacteroidetes and Clostridia were typing errors and removed.

5.     A new reference was added (87)

Table 1

14

Line 318 diversity is important, anything about ASD and bacteria diversity?

Improved by adding more information

355-379

15

Line 325 101 bacteria/gram? 1012?

Corrected to 101 bacteria/gram content in the stomach, to 1012 bacteria/gram in the colon

475

Round 2

Reviewer 3 Report

This manuscript has improved. More evidence of gut bacteria influence ASD were added to support the argument.

However, there is still inconsistence in gut microbiome difference between ASD and healthy children. Should discuss why this happened, and how can you still support the argument that bacteria can be the target in treating ASD.

  1. Line 80-83 “GM communicate with each other… and with ….host” how and reference? What do you mean?
  2. Line 98-100 reference?
  3. Line 288 desulfovibrio doesn’t produce propionice acid. But it has other significance as the major H2S producer. Does H2S affect ASD?
  4. Line 296 Clostridium perfringens
  5. Line 313-316, is this established that chronic constipation can be characterised by bifido and lacto decrease?
  6. Line 269-300 contradicting results of ASD children having less bacteroides first then saying a surge in propionic acid producers (which are mainly bacteroides in the gut) increase risk of ASD deterioration. Should admit there are different results from different studies, which lead to a discussion about why a well-designed study is needed.

Author Response

#

Reviewer Comment

Authors’ Response

Line number

1

Line 80-83 “GM communicate with each other… and with ….host” how and reference? What do you mean?

Added information

83-89

2

Line 98-100 reference?

Added reference

104-106

3

Line 288 desulfovibrio doesn’t produce propionic acid. But it has other significance as the major H2S producer. Does H2S affect ASD?

Desulfovibrio was removed from propionic acid producers. The revised statement is limited only to Bacteroides and Clostridia (some species)

325-326

H2S discussion added

304-313

Desulfovibrio and ASD discussion added

302-303, 317-324

4

Line 296 Clostridium perfringens

Corrected spelling

338

5

Line 313-316, is this established that chronic constipation can be characterized by bifido and lacto decrease?

Specified with adding references [90-92]

357

6

Line 269-300 contradicting results of ASD children having less bacteroides first then saying a surge in propionic acid producers (which are mainly bacteroides in the gut) increase risk of ASD deterioration. Should admit there are different results from different studies, which lead to a discussion about why a well-designed study is needed.

Added more information to discuss Bacteroides in context to ASD as well GID, reasons for different findings, limitations and justification for well-designed studies in the future. The paragraphs below further clarify our response

289-301, 329-334, 687-688

Bacteriodetes is a phylum that includes 241 genera (line 279) including Bacteroides. Some of the studies used in the preparation of this manuscript described the changes in GM at the phylum or genus levels, while a few other studies specified a particular species [e.g. in Table 1, Bacteroides fragilis was high (ref 105; this is a particular species) BUT the genera Bacteroides was low in ref 106]

We added a new reference (ref 74) of a recent systematic review on GM alterations in ASD. Among other genera, Bacteroides was increased in individuals with ASD relative to healthy controls in certain studies. We have added a paragraph to explain the causes of variation among different studies with notes on the altered genera based on the ref 47, as below (lines 289-301):  

"The heterogeneity in the results may be attributed to the variations in the methodological approaches used in different studies, such as sample sources, storage temperature, methods of DNA extraction, primers used in PCR, 16S rDNA sequencing platforms and data analyses methods. Other possible causes of heterogeneity can be related to the study participants’ characteristics including age, gender, type of control (sibling vs non-sibling), dietary factors, and GI symptoms. Thus, well-designed studies are needed for a better understanding of the significance of GM in ASD" (lines 294-301).